# COG6 is an essential host factor for influenza A virus infection

Daobin Feng,[1] Jiaohan Guo,[1] Jingjing Yan,[1] Jian Chen,[1] Longfei Ding,[1] Xinyu Zhu,[1] Zhilu Chen,[1] Yangyang Hu,[1] Miaomiao Zhang,[1] Jian Liu,[1] Cuisong Zhu,[1] Mingbin Liu,[1,2] Chen Zhao,[1] Xiaoyan Zhang,[1,3] Jianqing Xu[1,3]

**ABSTRACT** Influenza A virus (IAV) relies on the host cellular machinery to support its replication. Understanding these host dependencies can inform the development of novel antiviral strategies. In this study, we identified conserved oligomeric Golgi complex subunit 6 (COG6) as a novel host factor critical for IAV replication through a genome-wide clustered regularly interspaced short palindromic repeats/CRISPR-associated protein 9 (CRISPR/Cas9) knockout screen. Disruption of COG6 significantly impaired viral replication. Mechanistically, COG6 supports IAV replication via two distinct means. First, consistent with the role of the COG complex in Golgi homeostasis, COG6 is required for the proper presentation of surface sialic acids, the primary receptor for IAV entry. Second, COG6 deficiency unexpectedly led to lysosome-dependent degradation of viral proteins. Notably, lysosomal activity was also upregulated in IAV-infected wild-type cells, albeit to a lesser extent than in COG6-deficient cells. Treatment with lysosomal inhibitors rescued viral protein stability in COG6 knockout cells. Protein interaction analysis further demonstrated that COG6-mediated stabilization of viral proteins did not rely on viral protein-COG6 interaction, refuting the hypothesis that COG6 acts as a shield factor to protect viral protein from lysosomal degradation. Moreover, knockout of other COG subunits produced similar antiviral effects, suggesting that an intact COG complex is required for IAV replication. Together, these findings uncover a critical role of the COG complex in regulating IAV replication and highlight a previously unappreciated functional link between the Golgi and lysosomes that could be exploited for treating IAV infections.

**IMPORTANCE** Despite advances in virology, numerous host determinants facilitating influenza A virus (IAV) pathogenesis remain uncharacterized. Our study establishes conserved oligomeric Golgi complex subunit 6 (COG6) as a critical host factor promoting IAV infection through complementary mechanisms: receptor modulation and viral protein stabilization. This represents the first demonstration that the COG complex regulates viral pathogenesis through proteostasis mechanisms, fundamentally expanding our understanding of host-virus interactions at the organelle interface. These findings not only provide new perspectives on viral exploitation of Golgi trafficking networks but also identify potential therapeutic targets against evolving influenza strains.

**KEYWORDS** COG6, influenza A virus (IAV), host-virus interactions, lysosomal degradation, viral ribonucleoprotein (vRNP)

Influenza A virus (IAV), a member of the Orthomyxoviridae family, is a significant respiratory pathogen with a broad host range. Its high mutation rate contributes to both seasonal epidemics and occasional global pandemics, often resulting in severe health and economic consequences (1). Seasonal influenza affects approximately 10% to 20% of the global population annually, causing an estimated 290,000 to 650,000 deaths worldwide (1, 2). Historically, four major influenza pandemics occurred in 1918,

**Peer Reviewer** Kevin Ciminski, University of Freiburg, Freiburg, Germany

Address correspondence to Chen Zhao, chen_zhao72@163.com, Xiaoyan Zhang, zhangxiaoyan@fudan.edu.cn, or Jianqing Xu, xujianqing@fudan.edu.cn.

Daobin Feng and Jiaohan Guo contributed equally to this article. The author order was determined both alphabetically and in order of increasing seniority.

The authors declare no conflict of interest.

See the funding table on p. 16.

1957, 1968, and 2009 (3, 4). In 2024, the USA experienced a localized outbreak of highly pathogenic avian influenza, predominantly caused by the H5N1 subtype. This outbreak led to widespread infections in avian species, as well as spillover events affecting mammals such as cattle. Notably, a number of human cases were reported, highlighting the zoonotic threat and the potential for cross-species adaptation (5).

Ongoing antigenic drift and occasional antigenic shift continue to challenge the efficacy of current influenza vaccines and antiviral drugs. Although existing medical countermeasures, including neuraminidase inhibitors and polymerase inhibitors, can reduce disease severity and transmission, their effectiveness is often compromised by viral evolution (6–8). Consequently, there is an urgent need to develop next-generation, broadly protective vaccines and antivirals capable of counteracting diverse and emerging IAV strains.

IAV is an enveloped, single-stranded, negative-sense RNA virus with a segmented genome consisting of eight RNA segments. Its replication and transcription occur in the nucleus and are mediated by the viral ribonucleoprotein (vRNP) complex, which includes the RNA-dependent RNA polymerase (RdRp) and nucleoprotein (NP). The RdRp comprises three subunits (PB2, PB1, and PA), each of which contributes distinct enzymatic or structural functions essential for viral RNA synthesis.

Additionally, like other RNA viruses, IAV exploits host cell machinery to complete all stages of its replication cycle, including attachment, entry, genome release, nuclear import, transcription, replication, assembly, and egress. Numerous host factors are known to play critical roles in these processes and may serve as promising targets for host-directed antiviral strategies. However, the efficacy of available antivirals, including amantadine, oseltamivir, and baloxavir marboxil, is limited by resistance and narrow spectrum of activity (9, 10).

To deepen our understanding of virus-host interactions and identify novel therapeutic targets, numerous studies have employed high-throughput screening platforms. Among these, genome-wide clustered regularly interspaced short palindromic repeats/CRISPR-associated protein 9 (CRISPR/Cas9) knockout (KO) screens have proven to be particularly effective. This technology has led to the identification of multiple host genes essential for efficient IAV replication, including SLC35A1 (11), GNE (12), CMAS (13), CYTH2 (14), IGDCC4 (15), ATP6AP1 (16, 17), MYH9 (18), and components of the V-ATPase accessory complex, such as WDR7, CCDC115, and TMEM199 (19). Nevertheless, the intricate network of host factors involved in facilitating or restricting IAV replication remains only partially understood. Further in-depth studies are needed to elucidate these mechanisms and facilitate the development of innovative, host-targeted antiviral strategies.

In our study, we transduced the human GeCKO (genome-scale CRISPR knockout) lentiviral pooled library into A549 cells, followed by multiple rounds of infection with influenza A/Puerto Rico/8/1934(H1N1) virus. Surviving cells that exhibited resistance to infection were subsequently selected. We discovered that conserved oligomeric Golgi complex subunit 6 (COG6), encoding a subunit of the conserved oligomeric Golgi (COG) complex that is required for maintaining normal structure and activity of the Golgi apparatus (20), not only mediates cell surface expression of sialic acid receptors critical for viral entry, but also safeguards IAV proteins from lysosomal degradation while enhancing viral RNA replication and virion assembly. Notably, we demonstrated that nearly all other subunits of the COG complex exhibit similar capacity as COG6. These findings collectively establish a previously unrecognized mechanism whereby both COG6 and the holocomplex orchestrate multiple stages of the IAV life cycle.

## RESULTS

### Identification of host factors for IAV infection through genome-wide CRISPR/Cas9 knockout screening

To systematically identify host factors essential for IAV infection, we conducted a genome-wide CRISPR/Cas9 knockout screen with human GeCKO libraries. Shortly, A549 cells were transduced with human GeCKO lentiviral pooled libraries of 123,411 single-guide RNAs (sgRNAs) targeting 19,050 genes at multiplicity of infection (MOI) of 0.3 , followed by selection in puromycin for 14 days. Next, library cells were subjected to three rounds of stringent selection with PR8 virus at a high MOI of 5. Genomic DNA from surviving populations was sequenced and analyzed for sgRNA enrichment (Fig. 1A; Data S1). Bioinformatic analysis revealed top 10 candidate genes (Fig. 1B), among which polyclonal knockout lines targeting PCDHGA7, HSD17B14, LRRC43, COG6, TTC9C, TTC28, CLEC4C, and ANKHD1 were generated without compromising cellular viability (Fig. S1A). IFIT3 and TYK2, as known interferon-stimulated genes, were excluded from further validation. A subsequent IAV challenge demonstrated that only COG6 knockout significantly reduced viral infection by more than 50% compared to wild-type (WT) cells, prompting focused investigation of this component of the COG complex (Fig. 1C).

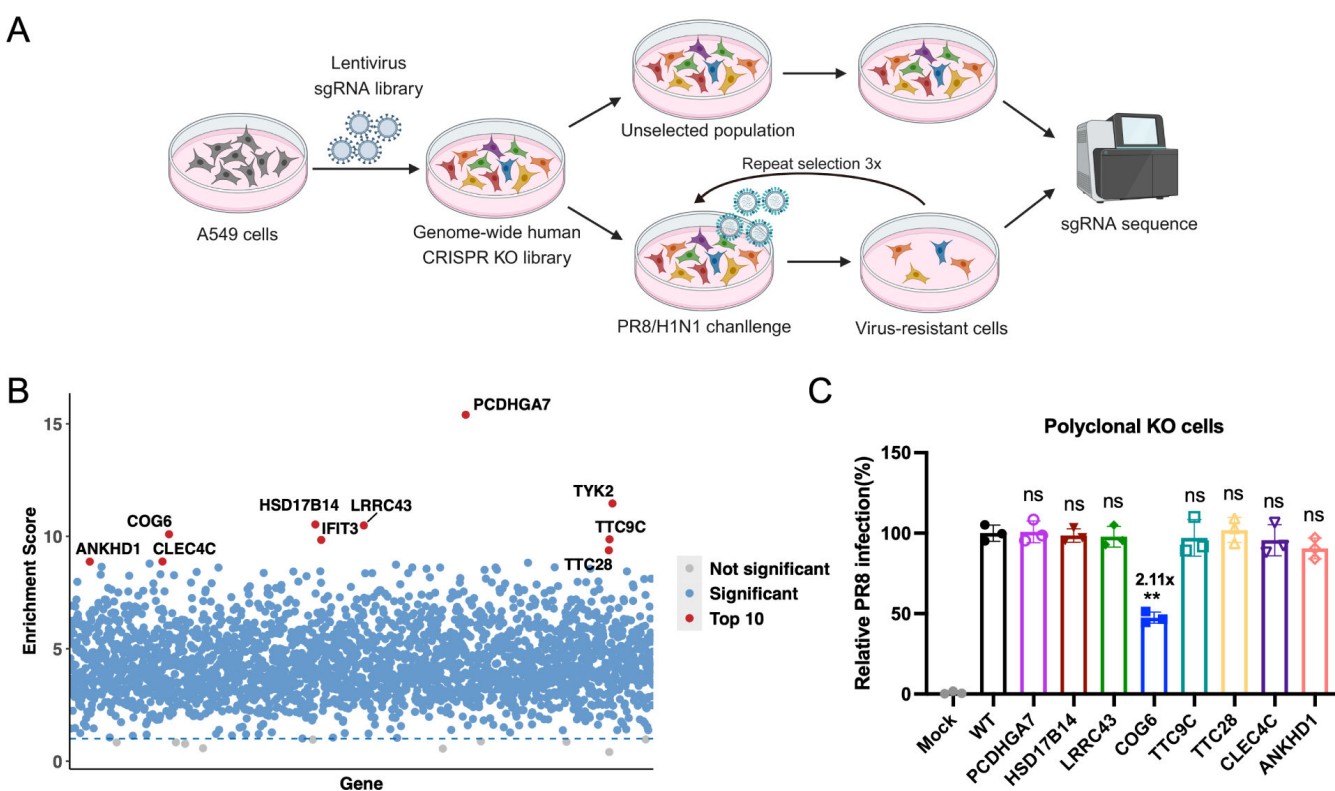

**FIG 1** CRISPR screening reveals host factor COG6 is required for IAV infection. (A) Flow chart showing the procedures of genome-wide CRISPR screening. (B) Manhattan plot of CRISPR screening results. The Manhattan plot displays the significance of each gene identified in CRISPR knockout screen. The $x$-axis represented the gene ID, while the $y$-axis showed the $-\log_{10}(P\text{-value}) \times \text{sign}(\log_2 FC)$ of the association between each gene knockout and infectivity. (C) Eight candidate gene knockout polyclonal cell lines were conducted by CRISPR/Cas9 technology (the sequences of sgRNA target site were shown in Table S1). These polyclonal cell lines were then infected with PR8 virus (MOI = 3). At 12 hours post infection (h.p.i.), IAV hemagglutinin (HA)-positive cells were detected using an Alexa Fluor 647-conjugated anti-HA antibody and analyzed by flow cytometry. Relative infection rates were normalized to those of WT cells. Data are presented as mean ± SD from three independent experiments. Statistical analyses were determined using one-way analysis of variance with Dunnett's multiple comparisons test.

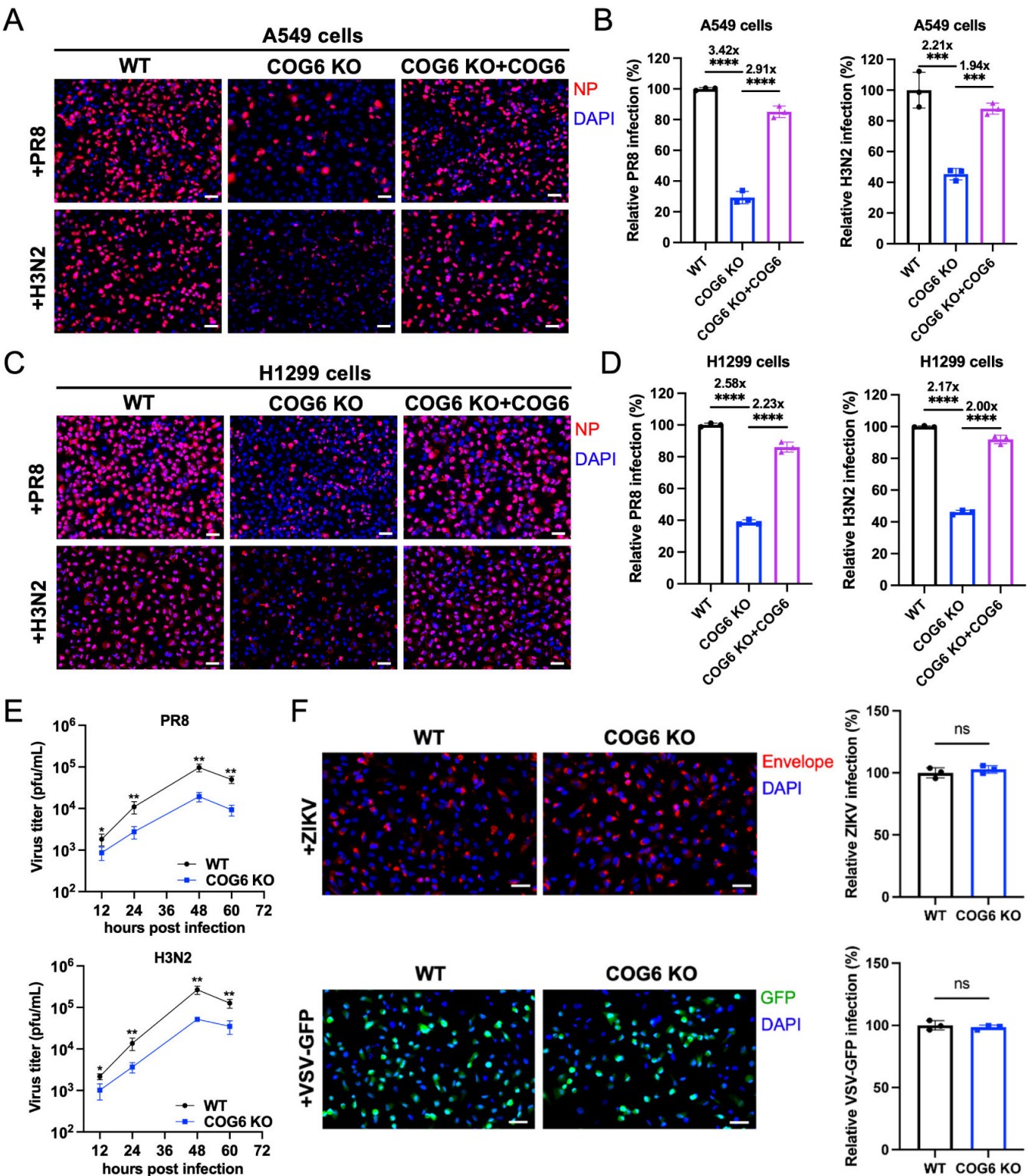

FIG 2  Knockout of COG6 suppresses IAV infection. (A and B) WT, COG6-KO, and COG6-complemented A549 cells were infected with PR8 or H3N2 virus at an MOI of 3. At 12 h.p.i., cells were fixed and stained for IAV NP in red and cell nuclei in blue (DAPI) and analyzed by TissueFAXS. Scale bars, 50 µm. The relative infection rates were quantified and plotted in (B). Over 1,000 cells were analyzed per biological replicate (*n* = 3). Infection rates were normalized to WT as 100% infection. (C and D) WT, COG6-KO, and COG6-complemented H1299 cells were infected and analyzed as described in (A and B). In (B and D), data are presented as mean ± SD from three independent experiments. (E) WT and COG6-KO H1299 cells were infected with PR8 (up) or H3N2 (down) virus at a low MOI of 0.01. At the indicated time points, supernatants were collected and virus titers were determined by 50% tissue culture infective dose assay in Madin-Darby canine kidney (Continued on next page)

Fig 2 (Continued)

cells. The values displayed are the $\log_{10}$ mean ± SD from three biological replicates. Some error bars are too small to be clearly visible. (F) WT and COG6-KO A549 cells were infected with ZIKV (MOI = 2) or VSV-GFP (MOI = 0.1). ZIKV-infected cells were fixed and stained for ZIKV envelope protein in red and nuclei in blue (DAPI) at 24 h.p.i., while VSV-GFP-infected cells were fixed and stained for VSV-GFP in green and nuclei in blue (DAPI) at 12 h.p.i. Scale bars, 50 µm. The relative infection rates were quantified and plotted. Over 1,000 cells were analyzed per biological replicate ($n = 3$). Infection rates were normalized to WT as 100% infection. Statistical analyses were determined using one-way analysis of variance with Dunnett's multiple comparisons test in (B and D) or unpaired, two-tailed Student's $t$-test in (E and F). *, $P < 0.05$; **, $P < 0.01$; ***, $P < 0.001$; ****, $P < 0.0001$; ns, no significance.

## COG6 enhances IAV replication

Monoclonal COG6-KO cell lines in A549 and H1299 cells (Fig. S1B and C) exhibited normal proliferation (Fig. S1D), yet displayed markedly reduced susceptibility to PR8 and H3N2 strains, with NP expression reduced to ~40% of WT levels by immunofluorescence (Fig. 2A through D). Viral titers in KO cells were consistently reduced by 0.5–1 log across multiple time points (12–60 h.p.i., MOI = 0.01; Fig. 2E). Complementation with Flag-tagged COG6 restored infectivity to near-WT levels (Fig. 2A through D; Fig. S1E). Specificity was confirmed by the unaltered infectivity of Zika virus (ZIKV) and vesicular stomatitis virus expressing GFP (VSV-GFP) in KO cells (Fig. 2F; Fig. S1F). These findings establish COG6 as a pan-cell-line dependency factor for IAV infection.

## COG6 facilitates IAV attachment

COG6 is a component of the COG complex, which is essential for maintaining the normal structure and activity of the Golgi apparatus. COG6 congenital disorder of glycosylation is caused by biallelic mutations in COG6, underscoring its critical role in glycosylation processes (21). Given that the early steps of influenza virus replication involve attachment to sialic acid receptors on the cell surface and subsequent internalization, we explored whether COG6 facilitated IAV attachment and internalization.

Briefly, WT and COG6-KO A549 cells were incubated with PR8 virus at an MOI of 10 at 4°C for 1 h (attachment assay) and then at 37°C for 45 min to allow viral entry (internalization assay). RT-qPCR quantification demonstrated that COG6-KO A549 cells exhibited significantly reduced PR8 virus attachment and internalization compared to WT cells (Fig. 3A). Immunofluorescence microscopy further confirmed that depletion of COG6 led to a marked reduction in IAV attachment and internalization (Fig. 3B).

We then detected α2,3- and α2,6-linked sialic acids on the cell surface using Maackia amurensis lectin (MAL) and Sambucus nigra lectin (SNA), respectively. Flow cytometry analysis revealed that depletion of COG6 resulted in a significant reduction in cell surface α2,3-linked sialic acids, while α2,6-linked sialic acids remained unaffected (Fig. 3C). These results indicate that COG6 is required for the proper expression of α2,3-linked sialic acids.

## COG6 stabilizes IAV protein expression

To assess whether COG6 is involved in additional stages of IAV infection, we investigated its role in viral RNA synthesis. WT and COG6-KO H1299 cells were infected with PR8 virus, and levels of NP vRNA, cRNA, and mRNA were quantified at 3 and 6 h.p.i. using RT-qPCR. In COG6-KO cells, all three RNA species were significantly reduced compared to WT cells at both time points (Fig. 4A).

To elucidate whether COG6 regulates viral RNA levels, we first examined its effect on IAV polymerase activity. WT or COG6-KO H1299 cells were co-transfected with vRNP components (PB2, PB1, PA, and NP) and a reporter plasmid encoding Firefly luciferase flanked by the 5′ and 3′ untranslated regions of the IAV NS segment. A Renilla luciferase-expressing plasmid was included as an internal control to normalize transfection efficiency. As shown in Fig. 4B, COG6 depletion significantly impaired viral polymerase activity, indicating that COG6 functions at a post-entry stage of infection.

Furthermore, COG6 knockout markedly decreased the expression levels of all vRNP components in the reconstituted IAV minigenome replicon system (Fig. 4C). Since the

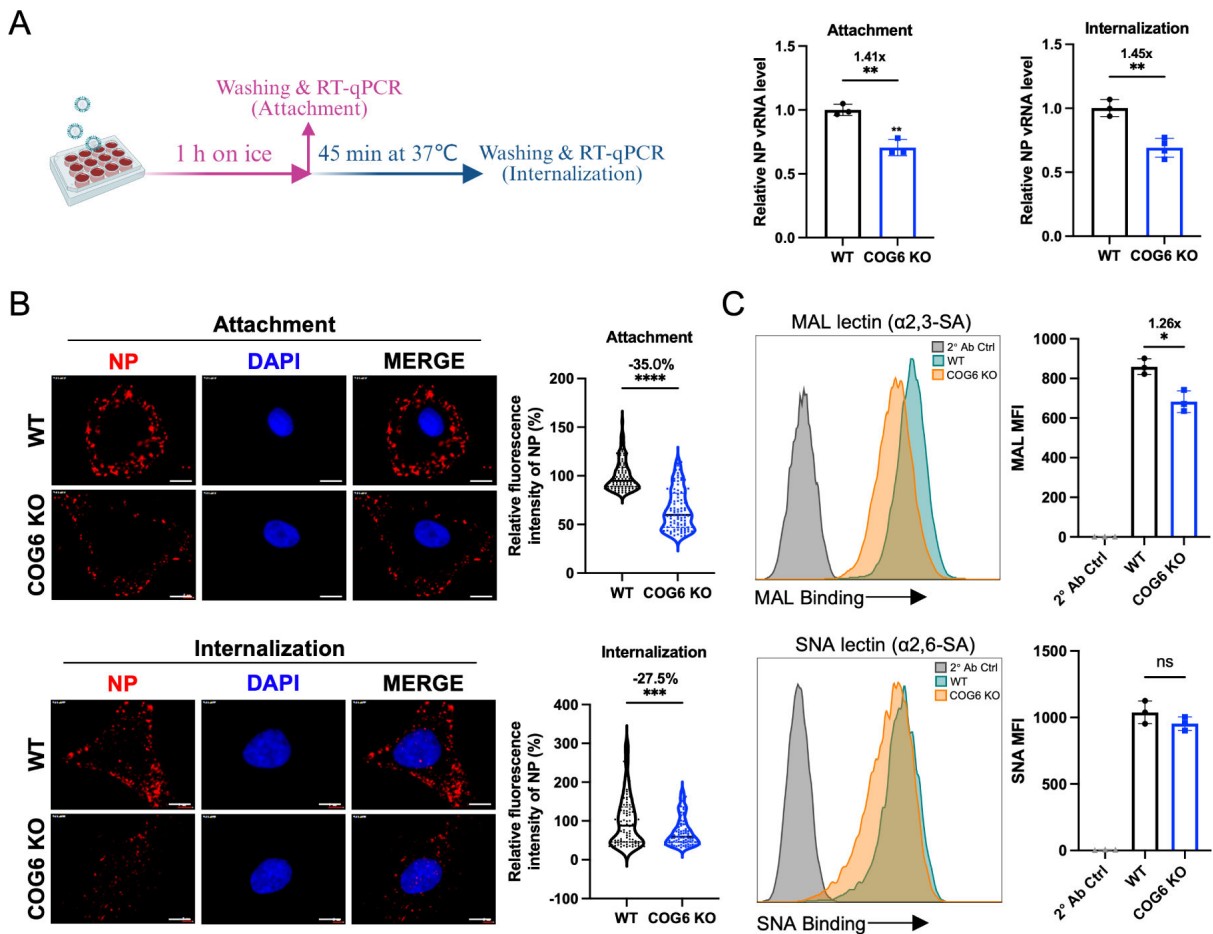

**FIG 3** COG6 promotes IAV attachment. (A) Evaluation of the effects of COG6 knockout on PR8 virus attachment and internalization. PR8 virus at an MOI of 10 was incubated with cells on ice for 1 h in the absence (attachment assay) or presence (internalization assay) of a follow-up incubation at 37°C for 45 min. Viral RNA of attached or internalized viruses was quantified by RT-qPCR and normalized to glyceraldehyde-3-phosphate dehydrogenase (GAPDH) mRNA levels. Data, presented as mean ± SD, are standardized to the corresponding RNA levels in WT cells from three independent biological replicates. (B) Cells described in (A) were infected with PR8 virus at an MOI of 10 and incubated with cells on ice for 1 h in the absence (attachment assay) or presence (internalization assay) of a follow-up incubation at 37°C for 45 min. At indicated time points, cells were fixed and stained for NP of PR8 virus in red and cell nuclei in blue (DAPI) and analyzed by TissueFAXS. Scale bars, 5 µm. The images were representative of three independent experiments. Quantification of the relative fluorescence intensity of NP was performed using Fiji software (WT, $n = 100$; COG6-KO, $n = 100$). The solid lines and the dashed lines represented median and quartiles of the data in the violin plots, respectively. Fluorescence intensity was normalized to background. (C) Surface levels of α2,3- and α2,6-linked sialic acid on WT and COG6-KO A549 cells were stained with biotinylated *Maackia amurensis* lectin (MAL) or *Sambucus nigra* lectin (SNA) lectin. Lectins were detected using phycoerythrin (PE)-labeled streptavidin. Representative histograms from $n = 3$ independent experiments are shown. Statistical analyses were performed by unpaired, two-tailed Student's *t*-test in (A–C). *, $P < 0.05$; **, $P < 0.01$; ***, $P < 0.001$; ****, $P < 0.0001$; ns, no significance.

expression of each viral gene in this system is driven by an exogenous plasmid promoter and does not rely on viral RNA polymerase activity, the observed reduction in viral protein levels in COG6-KO cells was not attributable to impaired viral transcription or replication.

To further examine the effect of COG6 on individual vRNP subunits, PB2, PB1, PA, and NP were expressed separately. As shown in Fig. 4D through G, COG6 deletion significantly reduced the protein levels of all four components. In contrast, the expression levels of ZIKV NS1 and severe acute respiratory syndrome coronavirus 2 (SARS-CoV-2) receptor-binding domain (RBD) were unaffected in COG6-KO cells relative to WT controls (Fig. 4H and I). These results suggest that in the absence of COG6, IAV proteins are targeted for degradation, indicating that COG6 contributes to the stabilization of viral proteins during infection.

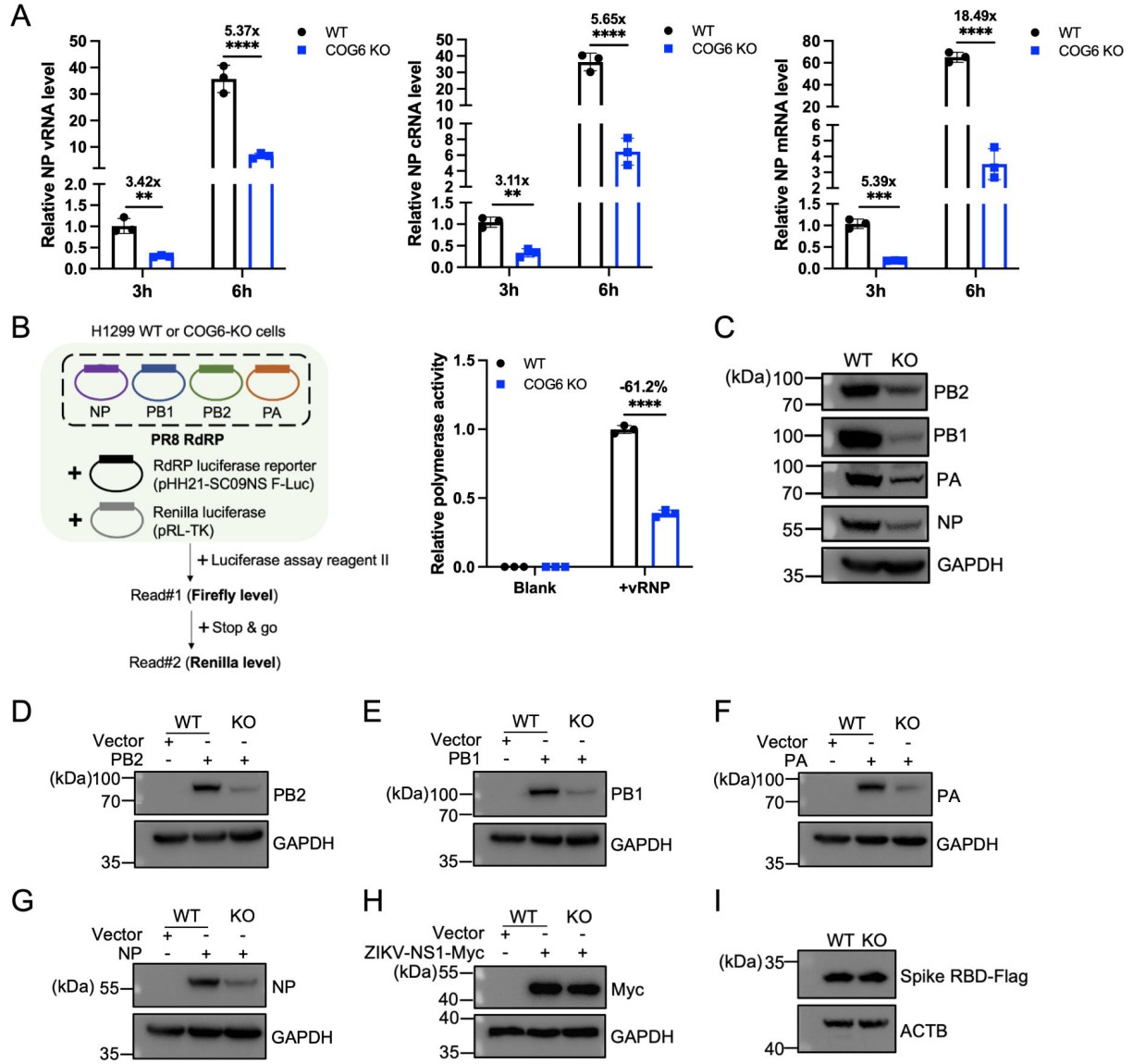

**FIG 4** COG6 stabilizes IAV proteins. (A) WT and COG6-KO A549 cells were infected with PR8 virus at an MOI of 3, and viral RNA species of NP of PR8 virus were quantified by RT-qPCR at indicated time points. Viral RNA species were normalized to GAPDH mRNA levels. Data, presented as mean ± SD, are standardized to the corresponding RNA levels in WT cells from three independent biological replicates. (B) A minigenome assay was conducted in WT and COG6-KO H1299 cells co-transfected with vRNP reconstitution plasmids (pCAGGS-PB2, pCAGGS-PB1, pCAGGS-PA, and pCAGGS-NP), a reporter construct plasmid containing negative-sense Firefly luciferase gene flanked by the 5′ and 3′ ends of IAV NS1 gene, and a plasmid expressing Renilla luciferase (internal control). Firefly and Renilla luciferase activities were measured at 24 h post-transfection. The Firefly luciferase values are normalized to the Renilla luciferase values. Data are presented as mean ± SD from three independent experiments. (C) The cell lysates from (B) were analyzed for protein expression of PB2, PB1, PA, and NP using western blot. GAPDH served as a loading control. (D–G) Western blot analyses were performed on lysates from WT and COG6-KO H1299 cells, each individually transfected with vRNP reconstitution plasmids including pCAGGS-PB2 (D), pCAGGS-PB1 (E), pCAGGS-PA (F), pCAGGS-NP (G), or the vector control. GAPDH served as a loading control. (H and I) Western blot analyses were performed on lysates from WT and COG6-KO H1299 cells transfected with a plasmid encoding ZIKV NS1 protein (H) or the receptor-binding domain (RBD) of severe acute respiratory syndrome coronavirus 2 (SARS-CoV-2) spike protein (I). GAPDH served as a loading control. Statistical analyses were determined using two-way analysis of variance with Sidak's multiple comparisons test in (A) or unpaired, two-tailed Student's *t*-test in (B). **, $P < 0.01$; ***, $P < 0.001$; ****, $P < 0.0001$.

## COG6 prevents IAV proteins from lysosomal degradation

The ubiquitin-proteasome system and the autophagy-lysosome pathway represent the two major protein degradation mechanisms in eukaryotic cells (22). To elucidate how COG6 stabilizes IAV proteins, WT and COG6-KO H1299 cells were individually transfected

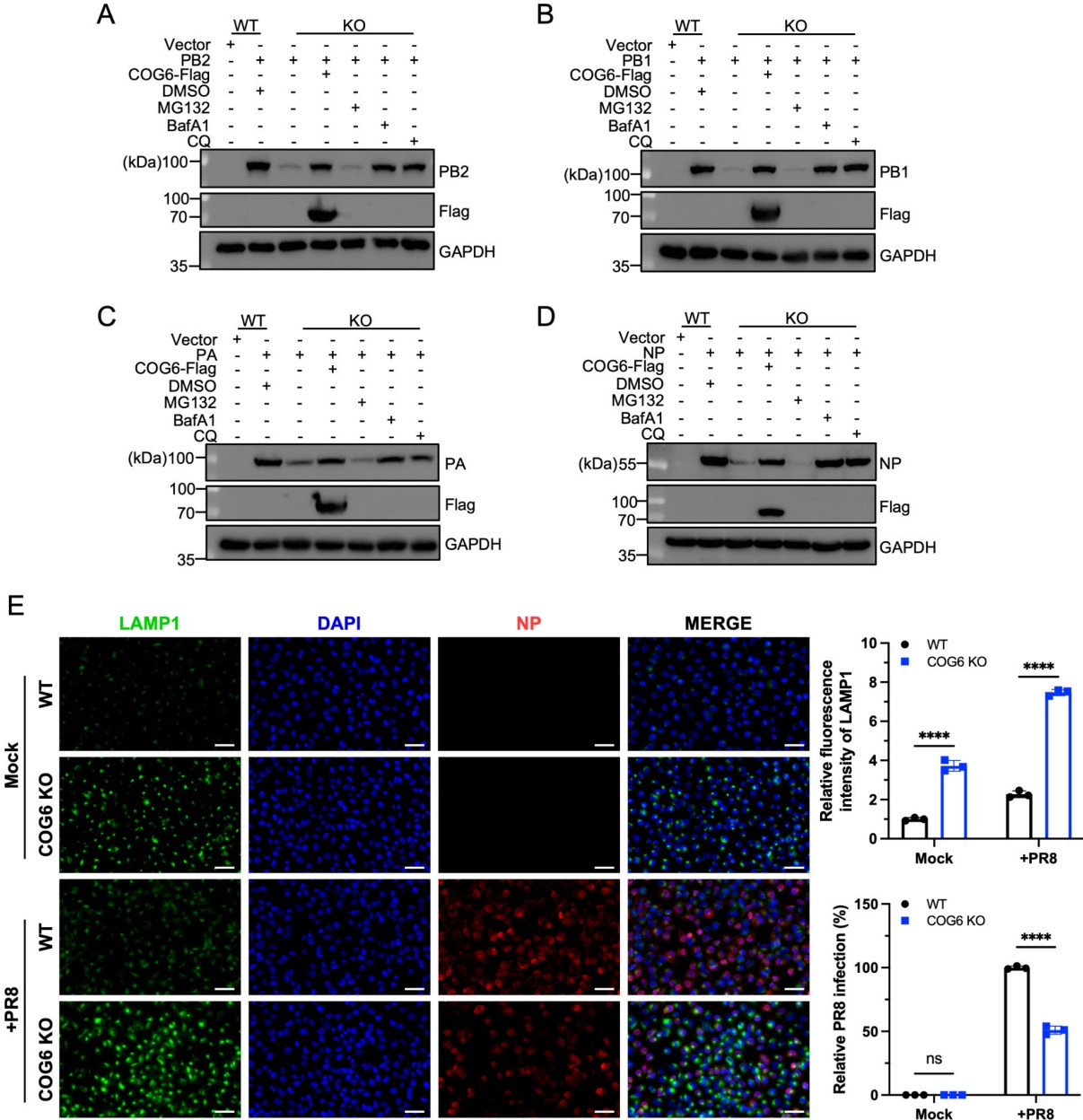

**FIG 5** COG6 prevents IAV proteins from lysosomal degradation. (A–D) Western blot analyses were performed on lysates from WT H1299 cells transfected with the vector control and COG6-KO H1299 cells transfected with vRNP reconstitution plasmids including pCAGGS-PB2 (A), pCAGGS-PB1 (B), pCAGGS-PA (C), or pCAGGS-NP (D). 24 h after plasmid transfection, the cells were treated with dimethyl sulfoxide, the proteasome inhibitor MG132, and the lysosomal inhibitors BafA1 or CQ. Additionally, COG6-KO H1299 cells were co-transfected with the pCAGGS-COG6 tagged with Flag (denoted as pCAGGS-COG6-Flag) and either pCAGGS-PB2, pCAGGS-PB1, pCAGGS-PA, or pCAGGS-NP. GAPDH served as a loading control. (E) WT and COG6-KO H1299 cells were mock- or infected with PR8 virus for 9 h at an MOI of 3. Cells were fixed and stained for LAMP1 in green, NP of PR8 virus in red, and nuclei in blue (DAPI). Scale bars, 50 μm. The relative fluorescence intensity of LAMP1 and the relative infection rates were analyzed by TissueFAXS. Over 1,000 cells were analyzed per biological replicate ($n = 3$). Infection rates were normalized to WT. Statistical analyses were determined using two-way analysis of variance with Sidak's multiple comparisons test. ****, $P < 0.0001$; ns, no significance.

with each vRNP component (PB2, PB1, PA, or NP), followed by treatment with the proteasome inhibitor MG132, lysosomal inhibitors bafilomycin A1 (BafA1), or chloroquine (CQ). As shown in Fig. 5A through D, treatment with BafA1 or CQ effectively prevented the degradation of vRNP proteins, consistent with the observation that complementation of COG6 in COG6-KO cells restored vRNP protein expression.

We next examined whether COG6 modulates lysosome biogenesis or function. In COG6-KO H1299 cells, we observed a marked increase in LAMP1-positive puncta, indicative of an elevated number of lysosomes, regardless of IAV infection (Fig. 5E). Collectively, these findings suggest that COG6 deficiency disrupts lysosomal homeostasis and enhances lysosomal activity by promoting lysosome biogenesis, which functionally facilitates the degradation of IAV proteins.

## COG6-mediated stabilization of IAV proteins does not rely on viral protein-COG6 interaction

To investigate the biological mechanism by which COG6 protects IAV proteins from lysosomal degradation, we tested its potential interactions with IAV proteins, including PB2, PB1, PA, and NP. Co-immunoprecipitation (Co-IP) assays were conducted in 293T cells co-transfected with COG6 and each of these viral proteins. Immunoblot analysis revealed a strong interaction between COG6 and NP, and no detectable interactions with PB2, PB1, or PA (Fig. 6A and D through F). Consistently, COG6 was also effectively pulled down by NP (Fig. 6B). Furthermore, the interaction between COG6 and NP was RNA-independent, as treatment of cell lysates with RNase A did not disrupt their association (Fig. 6C). Collectively, these findings demonstrate that COG6 robustly interacts with NP, while its interactions with other viral proteins are weak or undetectable, indicating that COG6 may not broadly bind to IAV proteins to prevent their lysosomal degradation.

## The COG complex facilitates IAV infection by promoting viral attachment and preventing viral proteins from lysosomal degradation

The COG complex is comprised of COG1, COG2, COG3, COG4, COG5, COG6, COG7, and COG8 subunits (23–25). Our study demonstrated that COG6 is a critical host factor that facilitates IAV infection by presenting sialic acid on the cell surface and protecting viral proteins from lysosomal degradation. To determine whether other subunits of the COG complex share this function, we generated seven knockout H1299 cell lines for

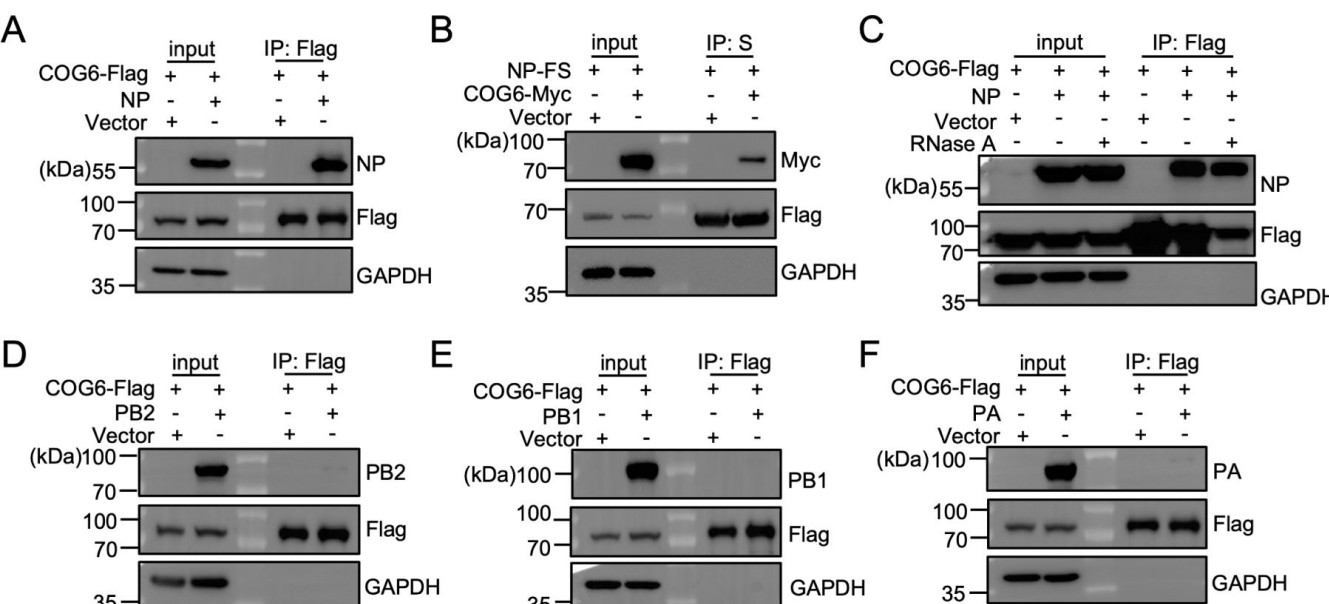

**FIG 6** COG6 interacts with NP. (A and D–F) 293T cells were co-transfected for 24 h with pCAGGS-COG6-Flag along with either the vector control, pCAGGS-NP (A), pCAGGS-PB2 (D), pCAGGS-PB1 (E), or pCAGGS-PA (F). Cell lysates were immunoprecipitated using anti-FLAG M2 affinity gel and analyzed by immunoblotting with the indicated antibodies. (B) 293T cells were co-transfected for 24 h with pCAGGS-COG6 tagged with Myc (denoted as pCAGGS-COG6-Myc) and either pCAGGS-NP tagged with Flag and S (denoted as pCAGGS-NP-FS) or the vector control. Cell lysates were immunoprecipitated using anti-S tag affinity gel and analyzed by immunoblotting with the indicated antibodies. (C) The cell lysates co-transfected with the pCAGGS-COG6-Flag and pCAGGS-NP were treated with RNase A (100 µg/mL) at 4°C for 1 h prior to immunoprecipitation with anti-FLAG M2 affinity gel. GAPDH served as a loading control.

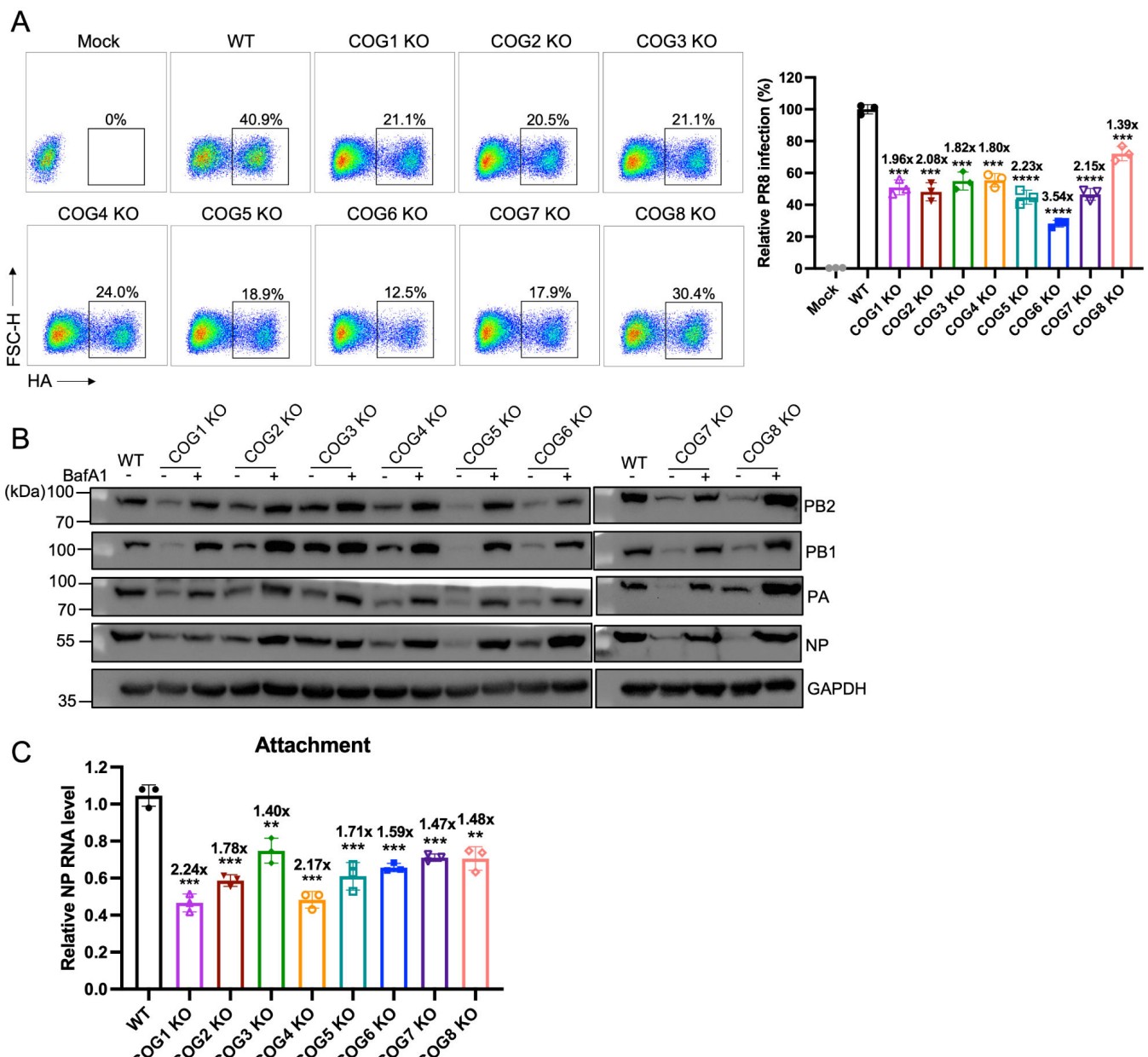

**FIG 7** The COG complex facilitates IAV infection by promoting viral attachment and protecting the viral proteins from lysosomal degradation. (A) Representative flow cytometric plots depicting frequencies of HA-positive cells from infected and uninfected cultures. WT, COG1-KO, COG2-KO, COG3-KO, COG4-KO, COG5-KO, COG6-KO, COG7-KO, and COG8-KO H1299 cells were infected with PR8 virus at an MOI of 3. At 12 h.p.i., cells were fixed, stained for IAV HA, and analyzed via flow cytometry. Data are presented as mean ± SD from three independent experiments. (B) Western blot analyses were performed on lysates from WT and COG6-KO H1299 cells transfected with vRNP reconstitution plasmids (pCAGGS-PB2, pCAGGS-PB1, pCAGGS-PA, and pCAGGS-NP) and COG6-KO H1299 cells following treatment with the lysosomal inhibitor BafA1. GAPDH served as a loading control. (C) Evaluation of the effects of WT, COG1-KO, COG2-KO, COG3-KO, COG4-KO, COG5-KO, COG6-KO, COG7-KO, and COG8-KO H1299 cells on PR8 attachment. PR8 virus at an MOI of 10 was incubated with cells on ice for 1 h. Viral RNA of attached PR8 virus was quantified by RT-qPCR and normalized to GAPDH mRNA levels. Data, presented as mean ± SD, are standardized to the corresponding RNA levels in WT cells from three independent biological replicates. Statistical analyses were determined using one-way analysis of variance with Dunnett's multiple comparisons test in (A and C). **, $P < 0.01$; ***, $P < 0.001$; ****, $P < 0.0001$.

COG1-KO, COG2-KO, COG3-KO, COG4-KO, COG5-KO, COG7-KO, and COG8-KO (Fig. S2A through G). These cell lines showed no significant impact on cell viability (Fig. S2H) and were subsequently infected with the PR8 virus. The results showed that knockout of any individual COG subunit significantly impaired PR8 virus infection (Fig. 7A).

To further explore this effect, COG1-KO, COG2-KO, COG3-KO, COG4-KO, COG5-KO, COG6-KO, COG7-KO, and COG8-KO H1299 cells were co-transfected with vRNP components (PB2, PB1, PA, and NP). Notably, the absence of nearly any subunit of the COG complex led to destabilization of vRNP proteins, resulting in their degradation via the lysosomal autophagy pathway (Fig. 7B). However, knockout of COG3 predominantly affected PA stability (Fig. 7B).

We next evaluated the role of COG subunits in viral binding. WT, COG1-KO, COG2-KO, COG3-KO, COG4-KO, COG5-KO, COG6-KO, COG7-KO, and COG8-KO H1299 cells were incubated with PR8 virus at an MOI of 10 at 4°C for 1 h for viral attachment. RT-qPCR analysis revealed that the deletion of any subunit of the COG complex significantly reduced viral attachment (Fig. 7C). Consistently, we used MAL and SNA lectins to assess the expression of α2,3- and α2,6-linked sialic acids on the cell surface. Flow cytometry analysis revealed that, similar to COG6-KO cells, knockout of COG1, COG2, COG4, COG5, COG7, or COG8 reduced the expression of α2,3-linked sialic acids on the cell surface. In contrast, COG3 knockout had minimal effect on either of the sialic acids, suggesting that COG3 may not regulate sialylation but instead affects other glycosylation modifications of membrane proteins (Fig. S3A and B).

Collectively, these findings underscore the critical role of the COG complex in maintaining host cell glycosylation and preventing lysosomal degradation of IAV proteins.

## DISCUSSION

In recent years, a variety of screening methods have been employed to identify host factors exploited by influenza virus (11, 16, 17, 19, 26–29). Despite these efforts, our understanding of the influenza-host interactions remained incomplete. In this study, we conducted a genome-wide CRISPR/Cas9 knockout screening to uncover host factors critical for IAV infection and identified COG6 as a previously unreported contributor to viral susceptibility. We subsequently demonstrated that IAV infection was affected by the absence of COG6 through dual mechanisms: reduced surface sialic acid levels, resulting in the inhibition of virus attachment, and destabilization of viral proteins due to enhanced lysosomal activities. We further showed that most, if not all, other components of the COG complex are required for IAV susceptibility in the same way as COG6. Together, these results firmly establish the COG complex as a new host factor regulating IAV infection.

Previous studies have implicated the role of the COG complex in the infections of other viruses. Depleting COG6 or COG7 impaired bovine herpesvirus-1 replication in Madin–Darby canine kidney (MDCK) cells (30). Silencing of multiple COG subunits (COG2–COG8) significantly reduced HIV-1 replication (31), while disruption of any subunit impaired orthopoxvirus infection (32). A general conception from studies is that the dependency of viral replication on the COG complex relies on the activity of the COG complex as the cellular machinery specializing in governing retrograde membrane trafficking, thus required for maintenance of Golgi architecture and, consequently, proper glycosylation of glycoproteins and glycolipids. The receptors for many viruses need glycosylation to support effective interaction with the virus for viral attachment and entry. This conception aligns with the first effect of COG6 knockout on IAV replication, reduced surface presentation of sialic acid—the generally regarded receptor for influenza viruses.

Our discovery of the second effect of COG6 on IAV replication stemmed from the observation that viral polymerase activity was significantly reduced in COG6 knockout cells compared to control cells. Further investigation revealed that the protein levels of all three viral polymerase subunits, alongside the NP protein, were greatly diminished. This reduction occurs independently of viral polymerase formation, as it was also observed when each viral protein was expressed individually. We further pinpointed the cellular mechanism involved by examining the rescuing effects of inhibitors of different protein degradation pathways on the COG6 knockout-mediated reduction of

viral protein expression. Lysosomal inhibitors, BafA1 and CQ, but not the proteasomal inhibitor MG132, could largely rescue the viral protein expression. Based on these results, we conclude that lysosome-mediated degradation accounts for the reduced IAV viral protein expression in the absence of COG6. This is unexpected, as the activity of the COG complex is typically associated with Golgi apparatus regulation.

A straightforward explanation for the link between COG6 deficiency and lysosomal degradation of IAV viral proteins is that COG6 directly participates in this process, possibly by interacting with viral proteins and protecting them from lysosomal degradation. However, our co-immunoprecipitation experiments did not support this hypothesis. Among the viral proteins tested—PB2, PB1, PA, and NP—only NP showed a strong interaction with COG6. This suggests that the COG complex likely influences viral protein stability through an indirect mechanism. Given the established connection between the COG complex and the Golgi apparatus, we propose that this mechanism involves disturbed Golgi homeostasis, which has been shown to enhance lysosomal activity (20, 33). Notably, we observed increased lysosomal activity in IAV-infected WT cells compared to mock-infected controls, with greater activity in COG6-KO cells. To our knowledge, there are few reports linking lysosomal regulation to viral protein expression during IAV infection. Our findings demonstrate that enhanced lysosomal degradation— such as that seen in COG6-deficient cells—can significantly inhibit viral replication, supporting lysosomal regulation as a potential therapeutic strategy against IAV infection.

There are still a couple of questions worthy of future investigation. Firstly, what are the determinants and host proteins responsible for targeting IAV proteins for lysosomal degradation? It is interesting to observe that the enhancement of lysosome-mediated degradation due to COG6 deficiency does not extend to the NS1 protein of Zika virus or the RBD of SARS-CoV-2, suggesting a selection process is at play. Secondly, how does Golgi stress translate into increased lysosomal activity? Identifying the signaling pathway and key regulatory molecules involved holds promise as new targets for developing small molecules to treat influenza viral infections. Answering these questions will help us advance our understanding of the intertwined cellular network determining the outcome of IAV infection, as highlighted by the dual regulatory role of COG6 in IAV infection revealed here.

## MATERIALS AND METHODS

### Cells

Human embryonic kidney (HEK) 293T cells, human lung carcinoma A549 cells, human lung carcinoma H1299 cells, and MDCK cells were purchased from the American Type Culture Collection. 293T, A549, and MDCK cells were cultured in Dulbecco's modified Eagle medium (10-013-CV, Corning) with 10% fetal bovine serum (FBS) (C04001-500, Vivacell) and 1% penicillin-streptomycin (PS, 100 IU/mL of penicillin and 100 µg/mL of streptomycin) (C100C5, NCM Biotech). H1299 cells were cultured in Roswell Park Memorial Institute 1640 (10-040-CV, Corning) with 10% FBS and 1% PS. All cells were maintained at 37°C in a 5% $CO_2$ humidified incubator.

### Antibodies

Antibodies used in the study included mouse anti-Flag M2 (F1804, Sigma), mouse anti-GAPDH (AC002, ABclonal), β-actin Rabbit mAb (AC026, ABclonal), rabbit anti-influenza A virus NP antibody (GTX125989, GeneTex), rabbit anti-influenza A virus PA antibody (GTX125932, GeneTex), rabbit anti-influenza A virus PB1 antibody (GTX125923, GeneTex), rabbit anti-influenza A virus PB2 antibody (GTX125926, GeneTex), Zika virus envelope protein antibody (GTX133314, GeneTex), S-Tag rabbit pAb (A24465, ABclonal), goat anti-rabbit IgG-HRP (B2615, Santa Cruz Biotechnology), goat anti-mouse IgG-HRP (31,430, Invitrogen), Alexa Fluor 568-conjugated donkey anti-mouse IgG (H+L) (ab175472, Abcam), and Alexa Fluor 488-conjugated donkey anti-rabbit IgG (H+L) (ab150077, Abcam).

## Plasmids and molecular cloning

To construct the mammalian transient expression vector pCAGGS encoding COG6 fused with 3×Flag tag at C-terminus, 3×Flag was added to the 3′ end of COG6 by polymerase chain reaction (PCR). The gene-targeting sgRNA from Table S1 was synthesized and cloned into lentiCRISPRv2 vector (Addgene, 52961). The open reading frames of NP, PB1, PB2, and PA derived from influenza virus A/Puerto Rico/8/1934 (PR8) were cloned into pCAGGS vectors. A reporter plasmid carrying Firefly luciferase in the negative sense flanked by 5′ and 3′ terminal ends of IAV NS and a control reporter plasmid carrying Renilla luciferase driven by a thymidine kinase promoter for constitutive expression were generated. All plasmid constructs were confirmed by Sanger sequencing.

## Generation of genome-wide CRISPR/Cas9 libraries and screening

A human GeCKO v.2.0 library containing 123,411 unique sgRNAs targeting 19,050 genes was obtained from Addgene (#1000000048). The sgRNA plasmid library was amplified, purified, and packaged into lentiviral particles in HEK293T cells via co-transfection with psPAX2 and pVSV-G at a 4:3:2 ratio using TransIT-LT1 Transfection Reagent (#2306, Mirus). At 36–48 h post-transfection, the viral supernatant was collected, clarified by centrifugation at 3,000 rpm–5,000 rpm for 30 min, concentrated, aliquoted, and stored at −80℃.

For CRISPR sgRNA screening, A549-GeCKO cell library was generated as previously described (34). Briefly, A549 cells were transduced with the sgRNA lentiviral library at a low MOI of 0.1–0.3 to ensure that most cells received a single sgRNA. Transduced cells were centrifuged at 2,000 rpm for 1 h. After 24 h, cells were selected with puromycin for approximately 2 weeks and subsequently pooled. To maintain a 300-fold coverage of the sgRNA library, sufficient cells were seeded and infected with PR8 virus at an MOI of 5, followed by incubation until nearly all cells were eliminated. The surviving cells were harvested and replated. After a 3 to 5 day recovery period, a second round of screening was performed in the same procedure. Surviving cells were expanded, and genomic DNA was extracted using QIAamp DNA Mini Kit (51306, Qiagen). Genomic DNA from uninfected, mutagenized cells was also extracted to serve as an unselected control.

sgRNA sequences were amplified from genomic DNA by PCR as previously described (35). PCR products containing sgRNA sequences were purified and subjected to next-generation sequencing using Illumina HiSeq 3000 platform. Sequencing data were processed by normalizing sgRNA read counts to counts per million total reads; counts fewer than five were adjusted to five to reduce noise from low-abundance reads. sgRNA-level $P$-values were calculated using Fisher's method. Log$_2$ fold change (log$_2$FC) values for each sgRNA were determined by comparing their distributions to those of control sgRNAs from the uninfected population. Finally, enrichment scores were calculated by integrating both statistical significance and expression change magnitude using the formula:

Enrichment score = $-\log_{10}(P\text{-value}) \times \text{sign}(\log_2\text{FC})$

And enrichment score plots were generated based on upregulated genes.

## Construction of A549 and H1299 knockout, complemented, and overexpression cell lines

For generation of KO cell lines, sgRNA was annealed into double-strand pair, phosphorylated by T4 polynucleotide kinase (EK0031, Thermo Scientific) and inserted into BsmBI-digested lentiCRISPRv2 by T4 DNA Ligase (EL0011, Thermo Scientific). Lenti-CRISPRv2 vectors carrying gene-targeting sgRNA or non-targeting control, psPAX2, and pMD2.G were co-transfected into 293T cells, and lentiviruses were collected 48 h post-transfection by filtering the supernatant of 293T with 0.22 μm filters and used to infect WT A549 or H1299 cells in the presence of polybrene (107689, Sigma). Transduced cells were treated with puromycin for 14 days and isolated by limiting dilution to generate monoclonal knockout cell lines. Knockout cells were validated by DNA sequencing or western blotting.

For the generation of complemented and overexpression cell lines, cDNA was synthesized from total RNA isolated from WT A549 cells using GoScript Reverse Transcriptase (Promega, A5003). The coding sequence of COG6, fused with either a Flag or Myc tag, was cloned into pCAGGS vector for transient expression or into pHAGE-CMV-IRES-Puro/Hygro for stable expression. The tagged COG6 constructs in pHAGE were packaged into lentiviral particles and used to infect WT A549 or H1299 cells. Stable cell lines were established through puromycin/Hygro selection. The expression of COG6 was confirmed by western blotting.

## Virus stock production and infection assays

Influenza A/Puerto Rico/8/1934 (H1N1) and influenza A/Aichi/2/1968 (H3N2) virus were grown in specific pathogen free chicken embryos for propagation. Virus titers were determined in MDCK cells using 50% tissue culture infective dose assay. For infection assays, plated cells were infected with PR8 or H3N2 virus in maintenance medium containing 1% FBS and 1 µg/mL tosyl phenylalanyl chloromethyl ketone-treated trypsin. Plated cells were infected with ZIKV or VSV-GFP in maintenance medium containing 2% FBS.

## Immunofluorescence staining

Cells were seeded in Millicell EZ SLIDES (PEZGS0816, Millipore) 12 h prior to infection. Plated cells were infected with IAV, ZIKV, or VSV-GFP in maintenance medium for 1 h at 37°C. Supernatants were replaced into serum-free medium for subsequent incubation. The slides were then blocked with phosphate-buffered saline (PBS) containing 1% bovine serum albumin and 2% donkey serum for 30 min at room temperature (RT) and processed to incubation of primary antibodies diluted in blocking buffer for 1 h at RT. After three-time washes with phosphate-buffered saline with tween-20 (PBST), the slides were incubated with secondary antibody diluted in blocking buffer for 30 min at RT. Finally, cells were stained with ProLong Gold Antifade with DAPI (P36931, Life Technologies) for another 1 min for cell nucleus positioning. Immunofluorescence images were visualized by TissueFAXS 200 flow-type tissue cytometer (TissueGnostics GmbH, Vienna, Austria).

## Virus replication

WT and COG6-KO A549/H1299 cells were infected with PR8 or H3N2 virus at an MOI of 0.01, and supernatants were collected at the indicated time points for viral titration in MDCK cells. Similarly, WT and COG6-KO A549 cells were infected with ZIKV (MOI = 0.5) or VSV-GFP (MOI = 0.01), and supernatants were collected at the indicated time points for titration in A549 cells. Viral titers were determined using the Reed-Muench method.

## Cell viability assay

Cell viability was determined by Cell Counting Kit-8 (CCK-8, A311-01, Vazyme). Briefly, cells were pre-seeded in opaque-walled 96-well plates. A total of 10 µL of CCK-8 solution was added into each well and incubated at 37°C for 4 h, followed by measurement of absorbance at 450 nm. Data were from three or six biological replicates.

## Western blotting

Cells were lysed using 4×Protein SDS-PAGE Loading Buffer (9173, Takara) and denatured at 95°C for 5 min. Protein samples were resolved by SDS-PAGE and transferred to polyvinylidene fluoride membranes (10600021, Cytiva). Membranes were blocked with 5% non-fat milk in PBST for 1 h at RT and incubated with 5% non-fat milk-diluted primary antibodies overnight at 4°C. Membranes bound with primary antibodies were washed thrice with PBST and incubated with 5% non-fat milk-diluted secondary antibodies for 1 h at RT. Membranes were washed thrice with PBST and visualized by a enhanced chemiluminescent substrate using Bio-Rad ChemiDoc MP Imaging System.

## Cell surface glycan staining

Biotinylated MAL II (B-1265-1) and SNA (B-1305-2) were purchased from VectorLabs. WT and COG6-KO A549 cells were washed with PBS twice and incubated with biotinylated lectins for 1 h on ice, followed by incubation with 1 µg/mL PE streptavidin (405203, BioLegend). The levels of lectin binding were determined by flow cytometry.

## RT-qPCR

To assess viral NP levels, total RNA from virus-infected cells was extracted using Direct-zol RNA Miniprep (R2052, Zymo Research). Relative amounts of NP vRNA, mRNA, and cRNA were quantified by RT-qPCR as previously described (18). RNA levels were normalized to GAPDH mRNA copies.

## Virus attachment and internalization assay

WT and COG6-KO H1299 cells pre-seeded in 12-well plates were washed five times by cold PBS and infected with PR8 virus (MOI = 10) for 1 h on ice. For the attachment assay, virus-bound cells were washed five times with ice-cold PBS (pH = 7.2) and lysed with RNAzol. For the internalization assay, virus-bound cells continued to an additional 45 min incubation at 37°C and, subsequently, cells were washed three times with ice-cold acidic PBS (pH = 1.5) and lysed with RNAzol. Viral RNA was quantified by RT-qPCR.

## Minigenome assays for IAV polymerase activity

In order to measure the activity of reconstituted vRNP, WT and COG6-KO H1299 cells were seeded into a 24-well plate at $2 \times 10^5$ cells per well 1 day prior to transfection. After 12 h, cells were co-transfected with pCAGGS expression plasmid encoding PB2 (0.1 µg), PB1 (0.1 µg), PA (0.1 µg), NP (0.1 µg), 0.1 µg of a reporter plasmid carrying Firefly luciferase in negative sense flanked by 5′ and 3′ terminal ends of IAV NS gene, and 0.02 µg of control reporter plasmid with Renilla luciferase as a transfection and toxicity control using TurboFect Transfection Reagent (R0533, Thermo Scientific). Cells were co-transfected with the same plasmids excluding pCAGGS-NP as a blank control. At 24 h post-transfection, cells were lysed, and the luminescence was measured with Dual-Luciferase Reporter Assay System (E1910, Promega) according to the manufacturer's instructions on GloMax 96 microplate luminometer (E6501, Promega). The levels of vRNP luminescence were normalized to those of Renilla luminescence.

## Co-immunoprecipitation assay

Co-IP assays were performed as previously described (36). For COG6 immunoprecipitating the PB2, PB1, PA, or NP protein of PR8 virus, pCAGGS-COG6-Flag was co-transfected to 293T cells pre-seeded in 10 cm dishes with either pCAGGS-PB2, PB1, PA, or NP, respectively, using TransIT-LT1 Transfection Reagent. Cells were washed thrice with prechilled PBS and lysed with Pierce IP Lysis Buffer (87787, Thermo Scientific) supplied with 1% (vol/vol) protease inhibitor cocktail (P8340, Sigma) for 30 min on ice. The lysates were incubated with protein A/G agarose (20422, Sigma-Aldrich) for 1 h at 4°C and were then subjected to centrifugation at 12,000 × $g$ for 10 min at 4°C. The supernatants were transferred to a new tube and incubated with anti-Flag M2 affinity gel (A2220, Sigma) overnight at 4°C. Subsequently, the agarose was washed thrice by prechilled wash buffer and incubated with 3×Flag peptide (F4799, Sigma) overnight at 4°C. The eluted supernatants were transferred into new 1.5 mL tubes, lysed with 4×Protein SDS-PAGE Loading Buffer, and inactivated at 95°C for 5 min. The samples were processed to SDS-PAGE and western blotting. For RNase A treatment followed by Co-IP, supernatants were collected and divided into two parts; one part was supplied with RNase A (EN0531, Thermo Scientific) at a final concentration of 100 µg/mL, while the other part was kept untreated. After rotation of the samples at 4°C for 1 h and centrifugation at 14,000 $g$ for 10 min, supernatants were harvested, and other experimental procedures

were performed as previously described. For viral NP proteins immunoprecipitating COG6, pCAGGS-NP-FS were co-transfected to 293T cells pre-seeded in 10 cm dishes with pCAGGS-COG6-Myc, followed by Co-IP assay as described above, and cell lysates were immunoprecipitated using anti-S tag affinity gel (69704, Millipore).

## Inhibitor treatments

WT and COG6-KO H1299 cells were pre-seeded in 12-well plates and transfected with pCAGGS-NP, PB1, PB2, or PA for 24 h. Cells were treated with MG132 (25 μM, HY-13259, MedChemExpress), BafA1 (100 nM, HY-100558, MedChemExpress), or CQ (50 μM, HY-17589A, MedChemExpress). Cells were lysed with 4×Protein SDS PAGE Loading Buffer, denatured at 95°C for 5 min, and analyzed by western blotting.

## Statistical analyses

All statistical analyses were performed with Prism 9.0 software (GraphPad). Quantitative data are shown as mean ± SD from at least three independent biological replicates. Statistical significance was analyzed using two-tailed unpaired $t$-test, one-way or two-way analysis of variance, followed by Sidak's or Dunnett's multiple comparison tests. For quantification of infected cells, at least 1,000 fluorescent cells were imaged and counted with a flow-type tissue cytometer. In accordance with current standard practice, $P$-values ≤0.05 were considered statistically significant and denoted with an asterisk. Non-significant values were denoted as ns.

## ACKNOWLEDGMENTS

This work was supported by the National Key R&D Program of China (2022YFC2604100, 2023YFC2605602) and the National Science Foundation of China (82072273).

## AUTHOR AFFILIATIONS

[1]Shanghai Public Health Clinical Center & Institutes of Biomedical Science, Shanghai Medical College, Fudan University, Shanghai, China
[2]Department of Epidemiology and Biostatistics, School of Public Health, Gannan Medical University, Ganzhou, China
[3]Clinical Center of Biotherapy, Zhongshan Hospital, Shanghai Medical College, Fudan University, Shanghai, China

## AUTHOR ORCIDs

Daobin Feng http://orcid.org/0000-0002-9519-9360
Jiaohan Guo http://orcid.org/0009-0001-8236-0469
Chen Zhao http://orcid.org/0000-0002-0718-7707
Xiaoyan Zhang http://orcid.org/0000-0002-3193-1401
Jianqing Xu http://orcid.org/0000-0003-0896-9273

## FUNDING

| Funder | Grant(s) | Author(s) |
| --- | --- | --- |
| National Key Research and Development Program of China | 2022YFC2604100 | Xiaoyan Zhang |
| National Key Research and Development Program of China | 2023YFC2605602 | Jianqing Xu |
| National Natural Science Foundation of China | 82072273 | Chen Zhao |

## AUTHOR CONTRIBUTIONS

Daobin Feng, Formal analysis, Investigation, Validation, Visualization, Writing – original draft, Writing – review and editing | Jiaohan Guo, Formal analysis, Investigation, Methodology, Validation, Visualization, Writing – original draft, Writing – review and editing | Jingjing Yan, Formal analysis, Investigation | Jian Chen, Methodology, Writing – review and editing | Longfei Ding, Resources | Xinyu Zhu, Methodology | Zhilu Chen, Methodology | Yangyang Hu, Methodology | Miaomiao Zhang, Resources | Jian Liu, Methodology | Cuisong Zhu, Methodology | Mingbin Liu, Investigation | Chen Zhao, Conceptualization, Methodology, Writing – review and editing | Xiaoyan Zhang, Conceptualization, Funding acquisition, Project administration | Jianqing Xu, Conceptualization, Funding acquisition, Writing – review and editing

## DATA AVAILABILITY

Sequences are available in the NCBI Sequence Read Archive under BioProject accession number PRJNA1306987.

## ADDITIONAL FILES

The following material is available online.

### Supplemental Material

**Data Set S1 (Spectrum01362-25-s0001.xlsx).** CRISPR screening.
**Supplemental Material (Spectrum01362-25-s0002.docx).** Figures S1 to S3 and Table S1.

### Open Peer Review

**PEER REVIEW HISTORY (review-history.pdf).** An accounting of the reviewer comments and feedback.

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
