## [Reviewer comments · Microbiology Spectrum]

Microbiology Spectrum

COG6 is an essential host factor for influenza A virus infection

Daobin Feng, Jiaohan Guo, Jingjing Yan, Jian Chen, Longfei Ding, Xinyu Zhu, Zhilu Chen, Yangyang Hu, Miaomiao Zhang, Jian Liu, Cuisong Zhu, Mingbin Liu, Chen Zhao, Xiaoyan Zhang, and Jianqing Xu

Corresponding Author(s): Jianqing Xu, Fudan University

Review Timeline:

Submission Date:	May 4, 2025
Editorial Decision:	June 18, 2025
Revision Received:	July 22, 2025
Accepted:	July 29, 2025

Editor: Mathilde Richard

Reviewer(s): Disclosure of reviewer identity is with reference to reviewer comments included in decision letter(s). The following individuals involved in review of your submission have agreed to reveal their identity: Kevin Ciminski (Reviewer #2)

Transaction Report:

DOI: <https://doi.org/10.1128/spectrum.01362-25>

Re: Spectrum01362-25 (*COG6 is an essential host factor for influenza A virus infection*)

Dear Prof. Jianqing Xu:

Thank you for the privilege of reviewing your work. Below you will find my comments, instructions from the Spectrum editorial office, and the reviewer comments.

Revision Guidelines

Sincerely,
Mathilde Richard
Editor
Microbiology Spectrum

Reviewer #1 (Comments for the Author):

In this manuscript, the authors Feng et al identify a novel host factor COG6 in the regulation of the influenza life cycle. The manuscript is well written and the conclusions are supported by the results presented. Overall, it is a very well rounded study that opens potentially new avenues of exploration in understanding IAV infection.

Reviewer #2 (Comments for the Author):

In their manuscript entitled "COG6 is an essential host factor for influenza A virus infection," Feng et al. use a genome-wide CRISPR screen to identify the protein COG6 as an important factor in infection with influenza A viruses (IAV). COG6 is the sixth subunit of the conserved oligomeric Golgi (COG) complex, which consists of eight subunits in total. Following the CRISPR screen, the authors investigate the function of COG6 in the IAV life cycle using various assays. By knocking out COG6 in human A549 and H1299 cells, the authors demonstrate that IAV replication, including polymerase activity, is significantly impaired, while Zika virus and VSV infectivity remains unchanged in these COG6 cells. At the cellular level, KO of COG6 leads to lower concentrations of IAV receptors, α 2,3-sialic acids (but not α 2,6-sialic acids), and increased LAMP1 expression, indicating an increased number of lysosomes. Due to impaired lysosome homeostasis, COG6 KO cells undergo rapid, lysosome-mediated degradation of IAV proteins but not of ZIKV NS1 or SARS-CoV-2 spike RBD. Overall, the manuscript is clearly written and well structured. The experiments are well chosen and support the authors' conclusions. Here are just a few minor comments:

1. The statement in lines 99-100 requires a reference because the structure and activity of the Golgi apparatus are not examined in the manuscript.
2. On line 102, the sentence should probably read, "nearly all other subunits."
3. Refer to Figure 2F on line 132.
4. Lines 153-155: The experiments show that expression of α -2,6-sialic acid is not impaired. Therefore, the second part of the sentence should be removed.
5. Lines 162-163: Since it is not clear at this point in the manuscript whether the low levels of NP vRNA, cRNA, and mRNA are also the result of lower internalization of the particles, the sentence should be reworded, e.g.: "To elucidate whether COG6 regulates viral RNA levels, we first examined its effect on IAV polymerase activity."
6. Lines 198-200: Since COG6 specifically protects IAV proteins from degradation, this should be emphasized. "To investigate the biological mechanism by which COG6 protects IAV proteins from lysosomal degradation, we tested its potential interactions with IAV proteins, including PB2, PB1, PA, and NP."
7. Lines 231-234: The authors should remove the statement regarding α -2,6-sialic acids because the results do not support it.
8. Regarding Figures 2A and 2F, why did the authors not quantify the virus-positive cells via flow cytometry, as was done in Figure 7A? A growth curve for ZIKV and VSV is needed to demonstrate that unchanged infectivity is associated with unchanged viral replication.
9. Regarding Figure 5E, NP staining is missing to identify infected cells.

Point-by-point response to reviewers

We thank the reviewers for their constructive and insightful comments. We have carefully revised the manuscript in accordance with all suggestions, and each point has been addressed in detail below. This includes performing additional experiments as requested. We hope that the revised manuscript now meets the criteria for publication in *Microbiology Spectrum*.

Reviewer #1 (Comments for the Author):

In this manuscript, the authors Feng et al identify a novel host factor COG6 in the regulation of the influenza life cycle. The manuscript is well written and the conclusions are supported by the results presented. Overall, it is a very well rounded study that opens potentially new avenues of exploration in understanding IAV infection.

Response:

We greatly appreciate the encouraging comments from the reviewer.

Reviewer #2 (Comments for the Author):

In their manuscript entitled "COG6 is an essential host factor for influenza A virus infection," Feng et al. use a genome-wide CRISPR screen to identify the protein COG6 as an important factor in infection with influenza A viruses (IAV). COG6 is the sixth subunit of the conserved oligomeric Golgi (COG) complex, which consists of eight subunits in total. Following the CRISPR screen, the authors investigate the function of COG6 in the IAV life cycle using various assays. By knocking out COG6 in human A549 and H1299 cells, the authors demonstrate that IAV replication, including polymerase activity, is significantly impaired, while Zika virus and VSV infectivity remains unchanged in these

COG6 cells. At the cellular level, KO of COG6 leads to lower concentrations of IAV receptors, α 2,3-sialic acids (but not α 2,6-sialic acids), and increased LAMP1 expression, indicating an increased number of lysosomes. Due to impaired lysosome homeostasis, COG6 KO cells undergo rapid, lysosome-mediated degradation of IAV proteins but not of ZIKV NS1 or SARS-CoV-2 spike RBD. Overall, the manuscript is clearly written and well structured. The experiments are well chosen and support the authors' conclusions.

Response:

We greatly appreciate the reviewer's encouraging comments and nice summary of our work.

Here are just a few minor comments:

1. The statement in lines 99-100 requires a reference because the structure and activity of the Golgi apparatus are not examined in the manuscript.

Response:

We thank the reviewer for this constructive advice. To address this point, we have added a reference to support the statement in question. The following citation, "Blackburn JB, D'Souza Z, Lupashin VV. 2019. Maintaining order: COG complex controls Golgi trafficking, processing, and sorting. FEBS Lett 593:2466-2487", has been added at line 100 of the revised manuscript.

2. On line 102, the sentence should probably read, "nearly all other subunits."

Response:

Thank you for your suggestion. We have rephrased the sentence in question as you

recommended. The revised sentence now reads: “Notably, we demonstrated that nearly all other subunits of the COG complex exhibit similar capacity as COG6.” (line 102, revised manuscript).

3. Refer to Figure 2F on line 132.

Response:

We thank the reviewer for pointing out this oversight. We have now referred to Figure 2F on line 131 of revised manuscript, as you recommended.

4. Lines 153-155: The experiments show that expression of α -2,6-sialic acid is not impaired. Therefore, the second part of the sentence should be removed.

Response:

The reviewer’s point is well taken. Accordingly, we have revised the sentence in question as follows to more accurately reflect our experimental results: “Flow cytometry analysis revealed that depletion of COG6 resulted in a significant reduction in cell surface α 2,3-linked sialic acids, while α 2,6-linked sialic acids remained unaffected (Fig. 3C). These results indicate that COG6 is required for the proper expression of α 2,3-linked sialic acids.” (lines 148-151, revised manuscript)

5. Lines 162-163: Since it is not clear at this point in the manuscript whether the low levels of NP vRNA, cRNA, and mRNA are also the result of lower internalization of the particles, the sentence should be reworded, e.g.: "To elucidate whether COG6 regulates viral RNA levels, we first examined its effect on IAV polymerase activity."

Response:

The reviewer's point is well taken. We have replaced the original sentence with the one you suggested, and this change can be found on lines 158-159 of the revised manuscript.

6. Lines 198-200: Since COG6 specifically protects IAV proteins from degradation, this should be emphasized. "To investigate the biological mechanism by which COG6 protects IAV proteins from lysosomal degradation, we tested its potential interactions with IAV proteins, including PB2, PB1, PA, and NP."

Response:

We thank the reviewer for this constructive advice. Following the reviewer's suggestion, we have now rephrased the sentence on lines 194-196 of the revised manuscript.

7. Lines 231-234: The authors should remove the statement regarding α -2,6-sialic acids because the results do not support it.

Response:

We thank the reviewer for this critical comment. As suggested, we have removed the statement regarding α 2,6-linked sialic acids, since our results do not support this conclusion. The revised text (lines 227-229) now reads, "Flow cytometry analysis revealed that, similar to COG6-KO cells, knockout of COG1, COG2, COG4, COG5, COG7 or COG8 reduced the expression of α 2,3-linked sialic acids on the cell surface." We have also corrected the related statement in the Results section, as noted in our response to comment #4.

8. Regarding Figures 2A and 2F, why did the authors not quantify the virus-positive cells via flow cytometry, as was done in Figure 7A? A growth curve for ZIKV and VSV is needed to demonstrate that unchanged infectivity is associated with unchanged viral

replication.

Response:

We thank the reviewer for this valuable comment. Both immunostaining (as shown in Figures 2A and 2F) and flow cytometry (as used in Figure 7A) are useful and complementary methods for assessing viral infection. While both approaches allow quantification of virus-positive cells—flow cytometry being more convenient in this regard—immunostaining provides critical spatial information on viral protein localization within cellular compartments.

We fully agree that evaluating viral replication is essential to support the conclusion that COG6 deficiency does not impact ZIKV or VSV infection. In response to the reviewer's suggestion, we have now performed viral growth curve analyses for ZIKV and VSV-GFP in COG6-KO and WT cells. The results, now included in the revised manuscript as Figure S1F and also provided below for the reviewer's convenience, demonstrate that COG6-KO cells support viral replication at levels comparable to WT cells.

FIG S1F WT and COG6-KO A549 cells were infected with ZIKV (MOI=0.5) or VSV-GFP (MOI=0.01). The supernatants were collected at indicated time points and virus titers were determined by TCID₅₀ assay in A549 cells. The values displayed are the log₁₀ mean ± SD from

three biological replicates. Some error bars are too small to be clearly visible. Statistical analyses were determined using unpaired, two-tailed Student's *t*-test. *ns*, no significance.

9. Regarding Figure 5E, NP staining is missing to identify infected cells.

Response:

The reviewer's point is well taken. In response, we have repeated the immunostaining experiment using the NP antibody alongside the LAMP1 antibody to clearly distinguish infected cells. The resulting images have been included in the revised manuscript as the new Figure 5E and are also provided below for the reviewer's convenience.

FIG 5E WT and COG6-KO H1299 cells were mock or infected with PR8 virus for 9 h at an MOI of 3. Cells were fixed and stained for LAMP1 in green, NP of PR8 virus in red, and nuclei in blue (DAPI). Scale bars, 50 μ m. The relative fluorescence intensity of LAMP1 and the relative infection rates were analyzed by TissueFAXS. Over 1,000 cells were analyzed per biological replicate ($n=3$). Infection rates were normalized to WT. Statistical analyses were determined using two-way ANOVA with Sidak's multiple comparisons test. ****, $p < 0.0001$; *ns*, no significance.

Re: Spectrum01362-25R1 (*COG6 is an essential host factor for influenza A virus infection*)

Dear Prof. Jianqing Xu:

I am pleased to announce to you that your manuscript has been accepted, and I am forwarding it to the ASM production staff for publication. Your paper will first be checked to make sure all elements meet the technical requirements. ASM staff will contact you if anything needs to be revised before copyediting and production can begin. Otherwise, you will be notified when your proofs are ready to be viewed.

Sincerely,
Mathilde Richard
Editor
Microbiology Spectrum